## Perspective

Implementation Science; Mental Health; Latin America; Community Participation; Epistemology

**Corresponding author:**
Felipe Agudelo-Hernández;
Email: afagudeloh81703@umanizales.edu.co

# Decolonizing mental health: Rethinking implementation science from the ground up

Felipe Agudelo-Hernández[1] [iD], Lina Valeria Cuadrado[2] and Andrés Camilo Delgado-Reyes[3]

[1]Department of Mental Health, Dirección Territorial de Salud de Caldas, Manizales, Colombia; [2]Universidad El Bosque, Colombia and [3]Psychology, Universidad de Manizales, Colombia

## Abstract

Implementation science plays a crucial role in effectively translating scientific knowledge into sustainable, evidence-based health practices. This perspective article focuses on some Latin American experiences, highlighting the limitations of applying methodologies developed in the Global North to settings marked by structural inequalities, economic constraints and cultural diversity. The included experiences examine a range of programs, such as the national Breast-feeding and Feeding Strategy, the evaluation of the Triple P-Positive Parenting Program in Chile and the community component of Mental Health Gap Action Programme in Colombia. Other contributions explore professional training initiatives and offer critical reflections on frameworks, such as the Consolidated Framework for Implementation Research and the Reach, Effectiveness, Adoption, Implementation and Maintenance. The reflections call for strengthening local capacities, fostering meaningful South–South and South–North collaborations, and advancing a context-sensitive, equity-oriented approach to implementation science that supports the development of more adaptive, effective and just health systems.

## Impact statement

This article critically analyzes the application of implementation science in global mental health, emphasizing Latin American contexts to show how frameworks developed in the Global North can inadvertently reproduce colonial dynamics and overlook the sociocultural realities of low- and middle-income countries. Adopting a decolonial perspective calls for reimagining implementation science to center local epistemologies, community sovereignty and authentic co-creation. These insights are vital for policymakers, researchers and practitioners aiming to design mental health interventions that are equitable, effective and culturally resonant. Ultimately, the work advocates dismantling power asymmetries and fostering more just, sustainable health systems that genuinely respond to the diverse needs, values and aspirations of the communities they serve.

## Social media summary

Implementation frameworks require decolonial adaptations focused on equity and community determination.

## Introduction

Global mental health (GMH) has developed over the past two decades as a prominent yet contested field of inquiry, policy and practice. Initially conceived as a response to the disproportionate burden of mental disorders in low- and middle-income countries (LMICs), it has been driven by the promise of scalable interventions, international funding and universal access to care. Yet GMH has increasingly been criticized for reproducing colonial dynamics, privileging biomedical models and Northern expertise and marginalizing local knowledge, agency and systems of care (Mills, 2014; Beresford and Rose, 2023).

Within this context, implementation science – intended to bridge the gap between research and practice – has become central, offering frameworks to translate evidence into global interventions (Betancourt and Chambers, 2016; Glasgow et al., 2019; Shelton et al., 2020). However, its dominant models, grounded in standardization, replication and metric-driven accountability, often reduce the complexity of social realities in the Global South. These risks reinforce what Mills (2014, p. 4) terms "psychotropic citizenship": a regime where access to care depends on alignment with Western psychiatric norms. These critiques extend beyond delivery

models to underlying assumptions about mental health, which is frequently defined as the absence of individual pathology, neglecting collective, spiritual or historical dimensions of suffering and healing (Le et al., 2022).

This article advances a decolonial critique of such paradigms. Implementation science is not a neutral technical tool, but a political terrain shaped by histories of epistemic violence and ongoing asymmetries in global health governance. While it can serve as a vehicle for change, this requires a fundamental reimagining: grounding it in equity, co-creation and community sovereignty; centering local epistemologies; and dismantling extractive research logics.

We, therefore, propose reframing implementation as a struggle over what counts as evidence, who is authorized to lead and how transformation is envisioned. Drawing on critiques from the Global South and emerging frameworks, we call for dialogical, situated and justice-oriented approaches. Only then can GMH move beyond colonial legacies and foster more plural, equitable and locally meaningful futures (Abimbola et al., 2021; The Lancet Global Health, 2025).

## Critical analysis of implementation through Latin American experiences

This article critically examines implementation science in GMH through selected Latin American experiences, using them to illustrate challenges, innovations and propose decolonial alternatives. Rather than a systematic review, it adopts an illustrative case selection approach, focusing on programs that highlight the complexities of adapting global interventions to local contexts and the region's rich scholarship on collective health and implementation science (Silva et al., 2023).

Cases were included based on several criteria: relevance to decolonial critique, explicitly or implicitly challenging Western-centric paradigms and colonial legacies. A Latin American focus, reflecting the authors' situated knowledge and the region's strong tradition of community-based mental health initiatives (Silva et al., 2023). Also, program diversity, encompassing national strategies, intervention evaluations and community-led initiatives. Finally, a demonstration of key theoretical concepts, including tensions in applying standardized frameworks, the importance of local adaptation and the centrality of community sovereignty.

### Justification of analytical relevance

The selected Latin American cases serve as empirical anchors to illustrate how decolonial principles can be operationalized in practice, without claiming to provide a comprehensive overview of the region's mental health initiatives. By focusing on carefully chosen examples, the article moves beyond abstract critique to show tangible possibilities for transformative implementation science. These cases highlight the importance of local knowledge, context and power dynamics, contributing to a broader call for more equitable and just approaches globally. Although extensive implementation research exists across LMICs, centering Latin America allows deeper engagement with historical and sociocultural nuances, offering insights applicable to wider Global South contexts.

## Theoretical framework

### Implementation science: Origins, frameworks and limitations

Implementation science, emerging in the early 2000s, aims to bridge the research–practice gap by building on health services research and evidence-based policy, with frameworks like Consolidated Framework for Implementation Research (CFIR) and Reach, Effectiveness, Adoption, Implementation and Maintenance (RE-AIM) providing standardized approaches across health systems (Damschroder et al., 2009). While effective in some contexts, these frameworks prioritize scale, fidelity and cost-effectiveness over contextual responsiveness, which in LMICs can lead to poorly adapted interventions, marginalization of local expertise and reinforcement of power imbalances between global funders, external researchers and national implementers (Villalobos Dintrans et al., 2019). Abimbola and Pai (2020) caution that implementation science risks technocratic solutionism, favoring "what works" over "what matters," a concern echoed in systematic reviews emphasizing persistent challenges in achieving contextual relevance and equity (Means et al., 2020).

Traditional frameworks reflect Western, positivist assumptions that often overlook Global South realities. Decolonial adaptations center equity, contextual resonance and community self-determination: CFIR adaptations interrogate epistemic origins, address colonial legacies and structural inequalities, analyze institutional power, value Indigenous knowledge and promote co-creation and epistemic justice. A decolonial RE-AIM reconceptualizes reach as equitable access, effectiveness as community-defined well-being, adoption as genuine local ownership, implementation as adaptive and culturally responsive and maintenance as long-term community self-determination. Participatory data collection and community-defined outcomes are key strategies, transforming implementation science into a vehicle for justice, cultural revitalization and transformative health solutions.

Research using the Exploration, Preparation, Implementation, Sustainment (EPIS) framework reinforces these critiques by highlighting the dynamic interplay between inner and outer contexts across implementation phases (Moullin et al., 2019). A study of complex suicide prevention interventions across diverse settings (Krishnamoorthy et al., 2025) shows how stigma, inter-organizational dynamics, resource constraints and community readiness shape implementation in ways standard models overlook. The study also identifies bridging factors – stakeholder relationships and partnerships – as crucial connectors determining intervention success or failure, underscoring the need for nuanced, context-sensitive approaches to implementation science in global settings.

### Regional perspectives: Latin American contributions to implementation science

While much criticism of dominant implementation frameworks stems from global-level analyses, recent Latin American research provides grounded insights into both the challenges and possibilities of adapting implementation science to diverse contexts. As dos Santos Treichel and Hernández (2023) notes, the field emerged in the Global North, and its tools – developed in resource-rich environments – often fall short in addressing the complex sociopolitical and infrastructural realities of Latin American health systems. These reflections align with broader decolonial agendas, advocating for relational, plural and embedded implementation logics rather

than the uncritical replication of externally designed protocols (dos Santos Treichel and Hernández, 2023). Examples such as the Triple P–Positive Parenting Program in Chile demonstrate the potential and the constraints of implementation in the region (Errázuriz et al., 2016).

### Lessons from Latin America for the global south

The focus on Latin American experiences extends beyond regional specificity, serving as a lens to explore broader challenges and opportunities for decolonizing mental health implementation across the Global South. Latin America's history of colonialism and ongoing struggles for epistemic and political sovereignty offer lessons relevant to other LMICs.

**Shared colonial legacies and health disparities.** Like parts of Africa, Asia and Oceania, Latin America's health systems reflect the imposition of Western biomedical models, marginalization of Indigenous healing practices and persistent inequities. Consequently, Latin America provides a transferable framework for analyzing and challenging power asymmetries in diverse Global South contexts.

**Innovations in community-led approaches.** The region has a rich tradition of community-driven health initiatives and social movements offering alternative models of care. Examples include the Tejido de Salud of the Nasa Indigenous community in Colombia and the community mental health center *La puerta siempre está abierta*, which prioritize collective well-being, cultural relevance and local agency (Agudelo-Hernández et al., 2024). These initiatives offer practical blueprints for other regions developing contextually grounded and sustainable services.

**Navigating global health agendas.** Implementation of programs like Mental Health Gap Action Programme (mhGAP) highlights tensions between global initiatives and local realities. Practices of "strategic compliance," where local actors adapt global protocols to fit local priorities, demonstrate the importance of recognizing local adaptation and innovation rather than narrowly emphasizing fidelity to standardized models. Without following the guidelines of Colombia's National Mental Health Policy, this strategy failed to implement the community component of mhGAP, which has been poorly implemented in the contexts for which it was designed.

In sum, Latin American case studies and theoretical reflections illustrate universal struggles and innovative responses within the decolonial global health project. By linking regional insights to global debates, they advance a more interconnected, plural and equitable vision for mental health across the Global South. Latin American scholarship underscores strengthening local epistemic infrastructures and fostering genuine South–South and South–North collaborations. Such efforts can reduce reliance on Northern-centric models and funding, promoting resilient and equitable health systems.

### Decolonial perspectives on mental health

To decolonize mental health is to unsettle the colonial legacies embedded in global knowledge production, funding flows, diagnostic classifications and care delivery systems. It requires interrogating how Western psychiatric knowledge became hegemonic and asking what forms of knowledge were excluded, delegitimized or rendered invisible in the process (Mills, 2014). Decoloniality thus entails more than cultural adaptation; it demands a fundamental rethinking of the political economy and epistemology of mental health.

Mad Studies offers one strand of this work by reframing distress not as illness but as political, relational and socially embedded. Beresford and Rose (2023) argue that it challenges psychiatric authority and centers the voices of those with lived experience, aligning with disability justice and epistemic reparations. Complementary approaches from the Global South emphasize revitalizing Indigenous healing systems, collective rituals and community-based responses as valid and vital alternatives to biomedical frameworks (Abimbola et al., 2021; dos Santos Treichel and Hernández, 2023).

Decolonizing implementation, therefore, implies recognizing communities not just as stakeholders but as theorists, designers and ethical-political agents. It involves reimagining implementation beyond "scaling what works" toward facilitating sovereignty, healing and justice. This shift requires dismantling the assumption of Northern epistemic superiority and embracing multiplicity as a foundation for transformation.

### Decolonizing implementation sciences

A decolonial approach to implementation science requires moving beyond dominant notions of technical neutrality toward methods that are contextualized, dialogic and grounded in local knowledge systems. Implementation must be situated not only geographically but also historically and politically, acknowledging the legacies of colonialism, structural adjustment and verticalized health agendas that continue to shape how communities perceive and access mental health care.

Central to this approach is dialogic implementation, which prioritizes reciprocal engagement over one-time consultation. Rather than extracting insights to serve predefined projects, it emphasizes ongoing relationships of mutual transformation. The Friendship Bench in Zimbabwe exemplifies this orientation: by mobilizing culturally resonant concepts such as kufungisisa ("thinking too much") to frame mental health concerns, it demonstrates how language itself can serve as a tool of epistemic reterritorialization (Gone, 2016).

Such an approach requires slowness, uncertainty and humility – qualities that often clash with donor timelines, scale-driven logics and the fetishization of evidence. Ultimately, implementing dialogically is not merely methodological but constitutes an ethical stance against epistemic domination.

### Critical review of current implementation frameworks

Mainstream implementation science often relies on frameworks such as CFIR, RE-AIM, EPIS and the Implementation Outcomes Framework. While these tools have provided structure and drawn attention to the operational dimensions of health delivery, they embed assumptions that health systems are rational, stable and apolitical; that interventions can be universally categorized and benchmarked; and that scale is the ultimate marker of success (Villalobos Dintrans et al., 2019).

Viewed through a decolonial lens, these assumptions are problematic: they obscure the contingency, negotiation and struggle that characterize implementation in LMIC contexts, and they risk sustaining a "success literature" that privileges interventions aligned with donor expectations and measurable outcomes, regardless of their fit with local priorities, values or relational ecologies of care. For implementation frameworks to contribute to justice, they must move beyond fidelity metrics and embrace plural logics of value –

developing tools that capture cultural meaning, narrative resonance and collective well-being, not only individual behavior change or cost-efficiency.

## Examples of collaborative, co-created and community-led approaches

The examples highlighted in Table 1 portray that co-creation is not about tokenistic inclusion but about shifting the locus of authority, enabling communities to define their own problems, priorities and solutions. They also challenge romanticized notions of "community," revealing it as a space of internal difference and contested meanings that must be actively engaged. What distinguishes these initiatives is their ethical grounding in autonomy,

**Table 1.** Summary of decolonial case studies

| Case study | Key focus/intervention | Decolonial relevance/lesson |
|---|---|---|
| Brazil's National Breastfeeding and Feeding Strategy (Melo et al., 2024) | Promote breastfeeding and healthy complementary feeding through a multisectoral approach | Even well-intentioned national programs benefit from a decolonial lens to ensure they are truly context-sensitive and empowering |
| La Puerta Siempre Está Abierta (Agudelo-Hernández et al., 2024) | Community mental health center. Prioritize collective well-being, cultural relevance and local agency | Implementation of the global community mental health guidelines, despite not following Colombian public policy parameters, achieving the implementation of the community component of mhGAP |
| Triple P-Positive Parenting Program in Chile (Errázuriz et al., 2016) | Adaptation and implementation of an evidence-based parenting program | Adaptation alone is insufficient; epistemic assumptions underlying interventions must be scrutinized to avoid reproducing forms of epistemic violence |
| PRIME program in Sehore district, Madhya Pradesh, India (Shidhaye et al., 2019) | Comprehensive mental healthcare plan integrating mental health services into primary care | Success is attributed to community engagement, capacity building and supportive structures, rather than technological innovation |
| Friendship Bench initiative in Zimbabwe (Gone, 2016). | Problem-solving therapy delivered by trained lay health workers (grandmothers) using culturally embedded notions of distress | Affirming Indigenous framings of suffering and healing, demonstrating strong clinical and social outcomes |
| Tejido de Salud of the Nasa Indigenous community in Colombia (Asociación de Cabildos Indígenas del Norte del Cauca, 2022) | Community-rooted model of care, offering an alternative to Western biomedical models | Paradigm shift from Western biomedical models, demonstrating the capacity of community-rooted models to overcome institutional barriers |

dignity and solidarity, reminding us that implementation science must be a vehicle for epistemic democracy.

This critical stance is crucial, as romanticizing community can obscure inherent power dynamics, diverse interests and potential for exclusion within what is often perceived as a monolithic entity. Participatory processes, while aiming for inclusivity, can inadvertently be co-opted by local elites or reinforce existing structural inequalities if not carefully managed. Therefore, fostering genuine and transformative participation requires a nuanced understanding of community heterogeneity, active engagement with internal conflicts and mechanisms to ensure equitable representation and benefit-sharing among all members, particularly the most marginalized. This approach moves beyond an idealized vision to one that acknowledges and navigates the complexities of real-world community dynamics.

To complement the discussion on how these strategies can contribute to the decolonization of implementation sciences, it is essential to highlight their contributions to the Global North's implementation processes, demonstrating how local actors can transcend established frameworks and thereby enhance the effectiveness of strategies. For instance, the implementation of the Triple P program in Chile (Errázuriz et al., 2016) not only adapted but also surpassed and improved upon the original approach by Sanders et al. (2014) by integrating Chilean cultural and social particularities. This demonstrates that local adaptation can significantly enrich and enhance the effectiveness of interventions, offering valuable insights for global implementation efforts.

Similarly, the Brazil Breastfeeding and Complementary Feeding Strategy (Melo et al., 2024) not only aligns with the RE-AIM framework but also transcends it by incorporating a collective health and human rights perspective. This approach overcomes the limitations of purely biomedical models and illustrates how implementation can serve as a vehicle for social justice. This is further evidenced by how Melo et al. (2022) achieved greater effectiveness compared to Rollins et al. (2016) by focusing on the promotion of breastfeeding and complementary feeding from a comprehensive and contextualized perspective, emphasizing the importance of local context in achieving public health goals.

Furthermore, it is crucial to highlight how other strategies, such as the Friendship Bench initiative in Zimbabwe (Gone, 2016) and the Tejido de Salud of the Nasa Indigenous community in Colombia (Asociación de Cabildos Indígenas del Norte del Cauca, 2022), harmonize with the principles outlined in Table 1. These initiatives offer care models that prioritize collective well-being, cultural relevance and local agency, effectively challenging Western paradigms and demonstrating the capacity of community-based models to overcome institutional and epistemic barriers.

From these reflections, five interrelated principles for decolonial implementation emerge. Epistemic reflexivity interrogates whose knowledge is mobilized and challenges assumptions of political neutrality (Table 2). Communal sovereignty shifts from consultation to co-governance. Pluriversal evidence legitimizes diverse epistemologies, including storytelling, spiritual practices and collective healing. Ecological ethics situates mental health relationally, socially and environmentally. Temporal dignity advocates slow, locally meaningful temporalities rather than funding-driven urgency. These principles are generative, positioning implementation as a space for political creativity, negotiation of knowledge and redistribution of power.

The potential for genuinely decolonial implementation is limited by global health's institutional and financial structures, which concentrate knowledge production, funding, authorship and priority-

**Table 2.** Five Principles for a decolonial implementation science

| Principle | Description | Implications for practice |
|---|---|---|
| 1. Epistemic reflexivity | Continuous interrogation of whose knowledge is being mobilized, and to what ends. Rejects framings of implementation as politically neutral or culturally static | - Actively question the assumptions and values embedded in implementation frameworks.<br>- Prioritize local knowledge systems and community-defined needs.<br>- Acknowledge and address power imbalances in research and practice. |
| 2. Communal sovereignty | Shift from consultation to co-governance, where communities lead in defining problems, methods and measures of success | - Establish genuine partnerships with communities, ensuring their leadership in all stages of the implementation process.<br>- Support community-led decision-making and resource allocation.<br>- Respect and protect community data and knowledge. |
| 3. Pluriversal evidence | Recognition of diverse epistemologies, including storytelling, spiritual practices, collective healing and oral history, as legitimate foundations for intervention | - Broaden the definition of "evidence" to include qualitative, narrative and arts-based forms of knowledge.<br>- Integrate local healing practices and cultural traditions into mental health interventions.<br>- Develop culturally grounded and contextually relevant outcome measures. |
| 4. Ecological ethics | Insistence that mental health must be understood as relational, embodied and situated within broader social and environmental systems | - Address the social determinants of mental health, such as poverty, discrimination and violence.<br>- Promote holistic approaches to well-being that consider the interconnectedness of individuals, communities and their environments.<br>- Advocate for policies that support social and environmental justice. |
| 5. Temporal dignity | Resistance to the urgency imposed by funding cycles, advocating instead for slow, cyclical and locally meaningful temporalities in processes of healing and decision-making | - Challenge unrealistic project timelines and advocate for long-term, sustainable funding.<br>- Allow for iterative and adaptive implementation processes that are responsive to community needs and rhythms.<br>- Prioritize building trust and relationships over rapid results. |

setting in the Global North. Even well-meaning reforms can reproduce hierarchies under the guise of capacity building or scaling up (Kumar et al., 2024). The co-optation of decolonial discourse – "elite capture" – risks depoliticizing radical ideas while maintaining business-as-usual under progressive branding (Krugman, 2023).

Elite capture is the process by which elites appropriate the objectives of a social movement, seeking to fulfill their own agendas rather than those of those fighting for real change (Platteau, 2004; Dutta, 2009; Khan and Thorp, 2010; Táíwò, 2022). In this sense, this concept suggests that identity-based strategies risk losing key elements of community action, such as solidarity and the defense of human rights, and becoming pre-established categories by elites who, in the case of "La puerta Siempre Está Abierta," are the health system. Achieving transformative participation requires shared authority, co-governance, equitable partnerships and community-led decision-making, supported by investments in local institutions, context-specific capacity building, community-led monitoring and recognition of Indigenous knowledge.

This logic highlights the operation of a community mental health center, which does not "comply" with the regulations guided by Colombian public policies, especially those related to financing or mental health hospitalization standards. This center does not have physical restraint protocols, and although the country lacks funding mechanisms for community-based organizations, it manages to promote support groups and anti-stigma campaigns led by people with lived experience, within an intercultural framework, where it particularly supports Indigenous and Afro-Colombian communities.

Tensions also arise between international funding and local autonomy. Donor-driven agendas often dictate what is fundable and implementable, undermining self-determination and perpetuating coloniality. In Colombia, health workers report that global protocols frequently conflict with local priorities and political histories, creating a paradox between fidelity to protocols and responsiveness to communities. MhGAP exemplifies this tension, often sidelining local knowledge and care practices (Krugman, 2023; Agudelo-Hernández and Giraldo-Álvarez, 2025).

Decolonial practice also requires avoiding romanticized notions of "the community," which is heterogeneous and marked by internal hierarchies, exclusions and contested interests. Participatory processes, if uncritically facilitated, can be co-opted by local elites or reinforce structural inequalities, burdening the most marginalized. Ethical decolonial practice fosters dialog, negotiation and conflict resolution, ensures voluntary participation, equitably shares benefits and transparently addresses risks. By engaging these complexities, implementation can move beyond idealized participation toward genuinely transformative and just outcomes, redistributing epistemic and political authority while accommodating plural, contested visions of health, care and justice.

## Conclusions

Implementation science, emerging in the early 2000s, aims to bridge research and practice, drawing on health services research and evidence-based policy. Frameworks like CFIR and RE-AIM were developed to structure and standardize implementation across health systems. While useful, they prioritize scale, fidelity and cost-effectiveness over contextual responsiveness, often resulting in poorly adapted interventions, marginalization of local expertise and reinforcement of power asymmetries in LMICs. This technocratic orientation risks privileging what works over what matters, a concern echoed by systematic reviews highlighting persistent challenges in contextual relevance and equity.

Traditional implementation frameworks do not adequately reflect the realities of the Global South. Decolonial adaptations center on equity, contextual resonance and community determination. For CFIR, this entails interrogating epistemic origins, addressing colonial legacies and structural inequalities, analyzing institutional power dynamics, valuing Indigenous knowledge and prioritizing co-creation and epistemic justice. A decolonial RE-AIM reframes reach as equitable access, effectiveness as community-defined well-being, adoption as genuine local ownership, implementation as culturally responsive and maintenance as long-term community self-determination. Participatory data collection and community-defined outcome measures transform implementation science into a tool for justice, cultural revitalization and transformative health solutions.

**Open peer review.** To view the open peer review materials for this article, please visit http://doi.org/10.1017/gmh.2025.10095.

**Author contribution.** FAH: Conceptualization; roles/writing – original draft; writing – review and editing. LVC: Conceptualization; roles/writing – original draft. ACDR: Conceptualization; writing – review and editing.

**Competing interests.** The authors declare none.

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
