## [Reviewer Report]

This is an excellent article that makes critically important points--and does so well and clearly. My only suggestion is to cite a bit more work in this area, especially when introducing concepts. For example, when you bring up “elite capture,” I expected to see at least one or two citations to some of the early 2000/2010’s work using this concept and then maybe something more recent like Olúfẹ́mi O. Táíwò’s book.

Thank you for this work to imagine and guide the field toward a decolonial implementation science, and I hope this thinking you’ve laid out so well helps create real change.

---

## [Reviewer Report]

The manuscript presents a critical and deeply needed reflection on the limits of implementation science in global mental health, especially from a decolonial perspective and situated in Latin American experiences, which is valuable. Below are some recommendations to strengthen it:

Case Selection Methodology: Although the article is framed as an overview, the methodology used to select the cases analyzed is not clearly explained. It would be valuable to include a brief description of the criteria for including experiences, as well as a justification of their representative or analytical relevance.

Articulation between the regional and the global: The focus on Latin America is a strength of the manuscript. However, we suggest delving deeper into how the lessons learned from these experiences can engage with debates and practices in other regions of the Global South. This could enrich the comparative dimension and strengthen the global applicability of the proposals.

Visual summary of the proposed principles: The proposal of five principles for a decolonial implementation science is powerful and original. It is recommended to include a figure or table that summarizes them, which would facilitate their understanding and application by readers of diverse profiles (researchers, policy makers, health professionals).

Greater specificity in operational proposals: The article presents a solid critique of traditional frameworks (CFIR, RE-AIM); however, it could benefit from more detailed examples of how these frameworks could be redesigned or adapted from a decolonial perspective. This would allow us to move from critique to the construction of concrete alternatives.

Clarity and argumentative structure: Although the text is well written and articulated, some paragraphs are long and could be divided to improve readability. It is also suggested that the conceptual density of certain sections be reviewed to ensure the message reaches an interdisciplinary audience clearly.

Avoiding romanticizing the community: The manuscript lucidly addresses the risks of idealizing “the community.” It is recommended to maintain and, if possible, expand this critical reflection, especially in relation to the internal tensions, inequalities, and conflicts that participatory processes can present.

Strengthening the ethical and political dimension: The critique of the co-optation of decolonial discourse by international actors is pertinent. It would be useful to delve deeper into how to avoid this instrumentalization and what mechanisms could ensure genuine and transformative participation of local actors.

---

## [Reviewer Report]

This manuscript explores the intersection of implementation science and global mental health through a decolonial lens. It is authored by researchers based in Latin America and presents an important premise: to reframe global mental health implementation science by critically examining colonial legacies and promoting more contextually grounded approaches. Despite its relevance and strong intent, the manuscript lacks conceptual clarity, methodological transparency, and comprehensive engagement with the literature. These limitations weaken its ability to meaningfully contribute to current debates and advancements in the field.

Major comments:

- The manuscript lacks a clearly articulated objective. It reads as a combination of a narrative review, commentary, and conceptual proposal, but does not commit fully to any of these formats. The conclusion frames the article as a call to action, yet the foundations for this call—either empirical or conceptual—are insufficiently developed.

- The methods section is vague. The authors reference included articles in the abstract but do not specify how these were selected, analyzed, or synthesized. If this is intended as a narrative review, criteria for inclusion and thematic organization should be made explicit.

- The authors selectively cite literature, omitting foundational and recent contributions in global implementation science, including:

- Betancourt, T. S. and D. A. Chambers (2016). “Optimizing an Era of Global Mental Health Implementation Science.” JAMA Psychiatry 73(2): 99-100.

- Shelton, R. C., M. Lee, L. E. Brotzman, L. Wolfenden, N. Nathan and M. L. Wainberg (2020). “What is dissemination and implementation science?: an introduction and opportunities to advance behavioral medicine and public health globally.” International journal of behavioral medicine 27(1): 3-20.

- Means, A. R., C. G. Kemp, M. C. Gwayi-Chore, S. Gimbel, C. Soi, K. Sherr, B. H. Wagenaar, J. N. Wasserheit and B. J. Weiner (2020). “Evaluating and optimizing the consolidated framework for implementation research (CFIR) for use in low- and middle-income countries: a systematic review.” Implement Sci 15(1): 17.

As well as Latin American scholarship such as Silva, A. A., G. P. Lopes, H. G. Claro, P. R. Menezes, O. Y. Tanaka and R. T. Onocko-Campos (2023). “Collective Health in Brazil and Implementation Science: Challenges and Potentialities.” Global Implementation Research and Applications 3(4): 340-354. and broader critiques from LMICs. These omissions result in a narrow and unbalanced portrayal of the field.

- While the manuscript gestures toward decolonial theory, it does not introduce or define core paradigms being challenged (e.g., biomedical, positivist). Nor does it provide concrete alternatives. References to Mad Studies, epistemic justice, and community-based care are not contextualized, and key sources (e.g., Rivera-Segarra et al. 2022) are either underutilized or absent.

- The paper does not offer a framework or criteria for what a decolonized approach to implementation science might entail. Nor does it explain how the highlighted community programs were selected, evaluated, or linked to implementation science practices.

- The manuscript does not adequately cover global content. Its literature base is limited primarily to selected critiques and does not include the full breadth of research from across LMICs or global implementation efforts. It omits major recent contributions and frameworks explicitly designed to address contextual adaptation and equity in LMICs. I am not suggesting these frameworks adequately address the issue, but I expected authors to engage with them given the scope and objective of the article.

- Although the authors are based in Latin America, the manuscript does not sufficiently situate regional insights within a global context. It misses opportunities to highlight how Latin American movements—such as Brazil’s psychiatric reform—have contributed to global thinking in community mental health and implementation.

I recommend an invitation to resubmit as a shorter Perspective or Viewpoint article. The current manuscript raises essential issues but lacks the clarity, methodological rigor, and scholarly depth. A shorter, more focused piece could better serve the authors’ intent and the journal’s readership.

---

## [Reviewer Report]

This is a very well-written article with clear signposting and a convincing argument.

To answer the authors’ questions, the review has done a good job of linking the global with the local, highlighting the shortcomings of implementation science and suggesting improvements that reflect a decolonial lens.

You mention that “If implementation frameworks are to serve justice, they must evolve beyond fidelity metrics and embrace plural logics of value. This requires tools that can account for cultural meaning, narrative resonance, and collective

wellbeing”

You also state, “Likewise, the community mental health centre La puerta siempre está abierta offers a powerful example of how local initiatives can navigate and even subvert restrictive policy frameworks. Despite national regulations that limit the formal establishment of non-coercive community mental health centers, this program operated effectively as a space of care without coercion, grounded in relational trust and collective wellbeing. Its success highlights the capacity of community-rooted models to overcome institutional barriers by creating practice-based infrastructures of care”

I would have liked to see more engagement with local initiatives to demonstrate the barriers, challenges, navigating strategies, and lessons learned. For now, the paper describes the initiatives with limited detail, and it does not explain what is distinct or specific about these local contexts, cultures, and local meanings, what kind of adaptation and steps were taken to meet local needs, and what the successful outcomes were.

The introduction mentions that there will be an examination of a range of programmes. I could not find sufficient critique for Brazil’s national Breastfeeding and Feeding Strategy (EAAB) and evaluation of the Triple P-Positive Parenting Program in Chile.

Please explain what the EPIS is.

You have proposed five interrelated principles; it would be helpful to expand on how they could be applied, for example, by using a case study from one of the examples you mentioned above and explaining the difference this could make to the outcome.

In the conclusion, you suggest four interconnected shifts in practice. I wonder why you proposed the five principles above and then provided additional suggestions rather than incorporating them.

Finally, it is essential to highlight that the benefits of applying the decolonisation framework will be on a global level, as both the GN and the GS will benefit, and that lessons, theories, and knowledge emerging from the GS could improve implementation science in the West/GN. For example, the case of COVID-19 demonstrates how successful management of the virus primarily came from LMICs.

---

## [Reviewer Report]

Thanks for addressing the comments by reviewers. The paper looks good.

one small comment: to address the repetition in using the romanticized notions of “the community” at four places in the article-